# Red Blood Cell Folate Likely Overestimated in Australian National Survey: Implications for Neural Tube Defect Risk

**DOI:** 10.3390/nu12051283

**Published:** 2020-05-01

**Authors:** Shannon E. Hunt, Merryn J. Netting, Thomas R. Sullivan, Karen P. Best, Lisa A. Houghton, Maria Makrides, Beverly S. Muhlhausler, Tim J. Green

**Affiliations:** 1School of Agriculture, Food and Wine, Faculty of Sciences, University of Adelaide, Adelaide 5005, South Australia, Australia; Shannon.Hunt@sahmri.com; 2Women and Kids Theme, South Australian Health and Medical Research Institute, Adelaide 5000, South Australia, Australia; Merryn.Netting@sahmri.com (M.J.N.); Thomas.Sullivan@sahmri.com (T.R.S.); karen.best@sahmri.com (K.P.B.); Maria.Makrides@sahmri.com (M.M.); Bev.Muhlhausler@csiro.au (B.S.M.); 3Adelaide Medical School, University of Adelaide, Adelaide 5005, South Australia, Australia; 4School of Public Health, University of Adelaide, Adelaide 5005, South Australia, Australia; 5Department of Human Nutrition, University of Otago, Dunedin 9016, New Zealand; lisa.houghton@otago.ac.nz; 6Nutrition and Health, Health and Biosecurity Business Unit, Commonwealth Scientific and Industrial Research Organisation, Adelaide 5000, South Australia, Australia

**Keywords:** red cell folate, microbiological assay, immunoassay, folate measurement, neural tube defects

## Abstract

In 2009, the Australian government mandated the addition of folic acid to bread flour to reduce the incidence of neural tube defects (NTD)-affected pregnancies. In 2011–2012, the Australian Health Measures Survey (AHMS) reported a mean red blood cell (RBC) folate in women of reproductive age (16–44 y) of 1647 nmol/L. Over 99% of women had an RBC folate ≥ 906 nmol/L, a concentration consistent with a very low risk of NTDs if a woman became pregnant. However, RBC folate was measured using an immunoassay, which is not a recommended method due to questionable accuracy. The microbiological assay is the preferred method for RBC folate measurement. To determine whether the immunoassay method may have led to spurious conclusions about the folate status of Australian women, we collected fasting blood samples from 74 healthy non-pregnant, non-lactating women (18–44 y) and measured RBC folate using both the immunoassay and microbiological methods. Mean RBC folate (95% confidence interval) concentration measured with the immunoassay method was 1735 (1666, 1804) nmol/L compared with 942 (887, 1012) nmol/L using the microbiological method. No woman had an RBC folate < 906 nmol/L using the immunoassay method, whereas 46% of women had an RBC folate < 906 nmol/L using the microbiological method. The NTD risk was estimated to be 0.06% using the immunoassay method and 0.14% using the microbiological method. RBC folate using AHMS survey may have underestimated NTD risk in Australian women.

## 1. Introduction

Neural tube defects (NTD), such as spina bifida and anencephaly, are caused by the failure of the neural tube to close normally, at around 28 days post-conception. In controlled trials, it has been shown that folic acid taken prior to conception and early pregnancy reduces the incidence of NTDs by up to 80% [1,2]. In 2005 the Australian National Health and Medical Research Council recommended that all women planning a pregnancy take 400 ug of folic acid daily prior to conception and until the end of the first trimester to prevent NTDs Around 40% of pregnancies are unplanned in Australia [3], and because NTDs occur before many women know they are pregnant, more than 80 countries have mandated the addition of folic acid to food staples, typically wheat flour. In 2009, the Australian Food Regulation Ministerial Council mandated the fortification of bread flour with folic acid [4]. This increased folic acid intakes of the population and reduced the incidence of NTDs, especially among teenagers and Indigenous women [5].

To achieve a maximal reduction in NTDs with folic acid fortification, it is generally accepted that achieving a red blood cell (RBC) folate ≥ 906 nmol/L among women of reproductive age is desirable [6,7]. In the nationally representative 2011-2012 Australian Health Measures Survey (AHMS) the mean RBC folate in women of reproductive age (16–44 y) was 1647 nmol/L (relative standard error of 0.8%) with only 1% of sample concentrations < 906 nmol/L, suggesting that women were well protected against NTDs if they were to become pregnant [8]. The AHMS RBC folate concentrations are much higher than the mean concentration of 1057 nmol/L reported for women in the US National Health and Examination Survey, where more foods are fortified with folic acid and supplement use is higher [9].

The microbiological assay is considered the gold standard for RBC folate measurement, especially in population-based studies [6,10], and was the method used in the US national survey [9]. However, in the AHMS, RBC folate was measured by an immunoassay using a folate-binding protein on an automated clinical analyser. Although immunoassays are suitable for measuring plasma/serum folate where the predominant form of folate is 5 methyltetrahydrofolate, their accuracy for RBC folate measurement has been reported to be poor [10,11,12,13]. This lack of accuracy has been attributed to matrix effects in red cells as well as different binding affinities of the folate-binding protein for the various forms of folate found in red blood cells.

Our aim in this paper was to investigate whether the immunoassay used in the AHMS led to erroneous conclusions about the folate status of Australian women following folic acid fortification and calculated risk of an NTD affected pregnancy. Here, we compare RBC folate concentrations measured in blood samples collected from women using the immunoassay (as used in the AHMS survey) and the gold standard microbiological assay method.

## 2. Materials and Methods

### 2.1. Participants

Seventy-four healthy female volunteers were recruited through newspaper advertisements, posters, social media and leaflet distribution within Adelaide, South Australia. Women were eligible if they were not pregnant or breastfeeding, and/or had not taken folic acid containing supplements in the four months prior. Ethical approval was obtained from the University of Adelaide Human Research Ethics Committee HREC-2016-151 and women gave written informed consent.

### 2.2. Procedures

Women attended a morning clinic after fasting since midnight. Venipuncture blood samples were collected into two evacuated containers containing EDTA (Becton Dickinson PTY, Macquarie Park, NSW). One tube was sent within 3 hours to a commercial lab for a full blood count, including hematocrit using an automated hematology analyzer (Clinpath, Adelaide, SA, Australia). For the immunoassay, whole blood was aliquoted in two cryovials. For the microbiological assay, whole blood was diluted 1:10 in 1% ascorbic acid (Sigma Aldrich, St Louis, MO, USA) incubated for 30 minutes at 37 °C and then aliquoted. The remaining whole blood was centrifuged at 3000 g for 10 minutes at 4 °C and the resulting plasma aliquoted. All samples were stored at -80 °C until analyzed. 

### 2.3. Measurement of Red Cell Folate

Immunoassay: Frozen whole blood samples were sent to SA Pathology (Adelaide) for whole blood folate determination using an Elecsys® Folate RBC kit (Roche Diagnostics International Ltd, Rotkreuz, Switzerland) on a Roche Modular E 801 Immunology Analyzer (Roche Diagnostics International Ltd, Rotkreuz, Switzerland) [14]. A detailed explanation of the principle of the assay is available online. This method was identical to that used in the AHMS.

Microbiological assay: Whole blood lysates and plasma samples were sent on dry ice to The University of Otago, Dunedin, New Zealand for folate determination using the microbiological assay [15,16]. Briefly, the microbiological assay was conducted using a 96-well plate with chloramphenicol-resistant Lactobacillus rhamnosus (ATCC 27773) (American Type Culture Collection, Manassas, VA, USA) and 5-methyltetrahydrofolate calibrator (Merck and Cie, Schaffhausen, Switzerland). External and internal controls were included on each microplate. Intra- and inter-assay variation was less than 10% for plasma folate and 14% for red cell folate.

### 2.4. Data Analyses

Assuming a standard deviation of 358 nmol/L for the immunoassay method [7], a sample size of 74 women allows for mean RBC folate concentration to be estimated with a sufficiently narrow 95% confidence interval (width of approximately ± 80 nmol/L). Red blood cell folate was determined by the equation used in the AHMS which corrects whole blood folate for hematocrit. This assumes the amount of folate present in plasma is negligible. RBC folate concentrations were found to be approximately normally distributed for both the immunoassay and the microbiological method. The association between the two measurement methods was assessed using linear regression, with agreement quantified using a Bland–Altman plot. Predicted NTD risk based on RBC folate was then calculated for each method using the equation in Daly et al. [7]. All statistical calculations were performed using Stata 15.0 (StataCorp, College Station, Texas, USA). 

## 3. Results

The median (inter-quartile range) age of the women was 27 (23, 34) years, 92% were of European ethnicity, 78% had an undergraduate degree or higher, and 66% had a normal BMI (18.5–24.9 kg/m^2^). Mean RBC folate (95% confidence interval) concentration measured with immunoassay method was almost twice that of the microbiological method; 1735 (1666, 1804) compared with 942 (887, 1012) nmol/L, respectively. The mean difference between the two methods was 793 nmol/L (724, 862); *p* < 0.001), indicating a high degree of systematic error. There was a poor correlation between RBC folate measured using the two methods (R^2^ = 0.16; *p* < 0.001) (Figure 1A). The Bland–Altman plot also showed a poor level of agreement between the two methods (Figure 1B), with the immunoassay method expected to produce concentrations between 198 and 1388 nmol/L larger than the microbiological method for 95% of women (equivalent to the mean difference ± 2 standard deviations). The large width of the 95% limits of agreement indicates a high degree of random error. Using the immunoassay method, no women had an RBC folate < 906, whereas 34 women (46%) had an RBC folate < 906 nmol/L using the microbiological method.

## 4. Discussion

We have shown that RBC folate concentrations among women of reproductive age in Australia may have been overestimated in the AHMS survey. The use of immunoassays to measure RBC folate have been particularly problematic because assays, particularly calibrators, are not harmonized among manufacturers, they have high lot-to-lot variation, and are of questionable accuracy. Although immunoassays are suitable for serum folate where folate is present primarily as 5-methyltetrahydrofolate, the folate binding protein used in these assays has different affinities for the many forms of folate present in in red blood cells. Colapinto et al. [11] undertook a method comparison to allow for adjustment of RBC folate concentrations in the Canadian Health Measures Survey measured using an Immulite 2000 immunoassay (Siemens Canada Limited) with the US National Health and Nutrition Examination Survey using the microbiological assay. They reported a mean percent difference of 24% between the microbiological assay and immunoassay methods (95% limits of agreement -26% to 75%) and a correlation of 0.67 between methods. This finding compares to the absolute mean difference of 793 nmol/L (95% limits of agreement 198 to 1388 nmol/L) and weaker correlation of 0.40 in our study. To our knowledge, only one other study compared the immunoassay used in the Elecsys® Folate RBC kit to other immunoassays. Golding placed himself on a folate deficient diet and measured RBC folate weekly until megaloblastic anemia appeared. RBC folate measured using the AHMS system consistently gave much higher concentrations than other immunoassay methods as well as the microbiological method [12]. Another important finding was that the difference in folate values obtained using the two methods was not merely systematic but also random, meaning that the folate values estimated in the AHMS could not be reliably converted to equivalent values that would be obtained using the microbiological assay.

The present study has a number of strengths including the measurement of folate by the methods in the same participant blood sample, removing any intra-individual variation in folate concentrations. Secondly, the immunoassay used, the Elecsys® Folate RBC kit, is identical to that used in the AHMS. Finally, we have confidence in the values obtained from the microbiological assay. The microbiologic assay used by the University of Otago was compared against target values from samples provided by the US Centre for Disease Control in a round robin comparison [10]. Values produced by Otago were 4.2% ± 9.6% higher for RBC folate than the US Centre for Disease Control target value.

There are limitations to our study. Although attaining a red blood cell folate concentration of 906 nmol/L is a threshold recommended by WHO and others, it does not represent a threshold for the maximum prevention of neural tube defects. There is debate around the optimal cutoff for NTD prevention. An RBC folate concentration of 906 nmol/L was the lower bound of the upper quintile of red blood cell folate in the Irish case–control study [7]. We stress that the model for the relationship between RBC folate and NTD risk is continuous and have provided NTD risk estimates using both the immunoassay and microbiological methods (that differ substantially). Any increase in RBC folate would be expected to decrease NTD risk. However, we accept that our estimates of NTD risk based on RBC folate should be interpreted with caution as they are based on data from a different population [7]. 

Another important consideration is that we used 5-methyltetrahydrofolate as a calibrator, which is consistent with current recommendations, instead of folic acid as was used in earlier studies. It has been suggested that 5-methyltetrahydrofolate gives RBC folate concentrations from the microbiological method that are ~20% lower [17]. As such a lower cutoff of > 748 nmol/L for RBC folate has been suggested for NTD protection when 5-methyltetrahydrofolate is used as the calibrator. Krider et al., using data from two studies in China, a large population-based study (*n* = 247831) and a dose response trial (*n* = 1194), estimated 822 nmol/L (calibrator adjusted) as a cutoff [18]. In our sample, 27% and 34% of women still had RBC folate concentrations below 748 and 822nmol/L, respectively. Importantly, adjusting RBC folate measurements according to the calibrator used would not change conclusions about the high degree of systematic and random error of the immunological method. 

We are not suggesting that our sample is representative of the general Australian population, or that folic acid fortification has been ineffective in Australia. Clearly folic acid intakes have increased and rates of NTDs have decreased, especially among vulnerable population subgroups in vulnerable populations, as a result of the fortification [5]. Our mean RBC folate concentrations of 942 nmol/L are more consistent with the 1060 nmol/L reported in population-based US data (National Health and Nutrition Examination Survey 2008–2010) using the microbiological method [9]. 

## 5. Conclusions

In conclusion, there is an urgent need to assess the impact of folic acid fortification on RBC folate using accurate methods not only in women of reproductive age but also in non-target populations who are being exposed to folic acid with no known benefit such as children, men, and older people. Given that most clinical laboratories in Australia measure RBC folate using immunoassays, caution is warranted when interpreting these results.

## Figures and Tables

**Figure 1 nutrients-12-01283-f001:**
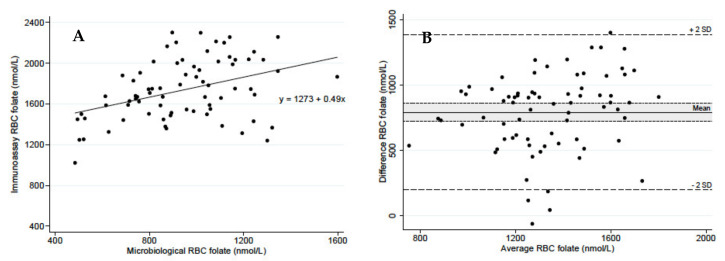
Comparison of red blood cell folate concentrations measured in concurrent venous blood samples using microbiological [chloramphenicol-resistant L. rhamnosus (ATCC 27773 or NCIB 10463) assay and erythrocyte folate measured by the protein-binding assay (Roche Modular E 801 Immunology Analyzer) (**A**) by regression; (**B**) and by a Bland–Altman Plot (difference on the y-axis is the immunoassay – microbiological RBC folate concentrations, with the shaded area corresponding to a 95% confidence interval for the mean difference).

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
