# Peer review of "Red Blood Cell Folate Likely Overestimated in Australian National Survey: Implications for Neural Tube Defect Risk"

_nutrients, 2020, doi:10.3390/nu12051283_

Round 1

Reviewer 1 Report

nutrients-767550-peer-review-v1

Shannon E. Hunt et al.

Red Blood Cell Folate Likely Overestimated in Australian National Survey: Implications for Neural Tube Defect Risk

This paper aimed to investigate whether the immunoassay used in the AHMS led to erroneous conclusions about the folate status of Australian people. The authors compared two RBC folate measurements, the immunoassay, and the microbiological assay. They found that the mean RBC folate concentration measured using the immunoassay was 793 nmol/L higher than that measured by the microbiological assay. Furthermore, this difference was random.  The authors compared the measurements of folate using the same participant blood sample and showed clear differences in results.

Major comment:

(1) Introduction part: insufficient background and relevant reference.

Brief background information on the following points is required.

  1. Why the accuracy of an immunoassay is poor? Please explain the principle of the assay. Are there multiple sub-methods in immunoassays? If so, please describe the difference between them and specify which type the AHMS used.
  2. How many sub-methods exist in microbiological assays? Is there any systematic bias among them? Are they compatible with each other? If so, how the threshold can be determined?

(2) Discussion part: related to the Introduction part, this part should be improved.

Although the microbiological assay can be recommended compared to the immunoassay, different microbiological assays could also give different results, depending on the folate calibrator and microorganism used. (PMID: 21613453) Daly study, from which the threshold value is determined, used folic acid as calibrator.  The authors used 5-methyltetrahydrofolate. Some conversion is required according to PMID:25073783? The authors should consider how to determine the threshold more carefully and provide a better solution.

Minor comment:

Reference 6, typo.

Author Response

We thank this reviewer for their excellent points, especially around the various iterations of the microbiological assay and different immunological assays.

The reviewer would like us to provide more details in the introduction around the immuno-assay and the many sub methods for the microbiological assay.

We have taken on board these comments, but to address them all in the introduction would lengthen the introduction of this Commentary. Instead we have addressed these comments as outlined below in various sections of the manuscript

Why the accuracy of an immunoassay is poor?

The references cited give the main reasons for poor accuracy in red cells (10-13).

We have given the main reasons for poor accuracy starting at ‘line 65.’

Although immunoassays are suitable for measuring plasma/serum folate, where the predominant form of folate is 5 methyltetrahydrofolate, their accuracy for RBC folate measurement has been reported to be poor [10-13]. This lack of accuracy has been attributed to matrix effects in red cells as well as different binding affinities of the folate-binding protein for the various forms of folate found in red blood cells.

Please explain the principle of the assay.

A full description of the method which is similar for all immunoassays can be found in reference 14. We have now added a URL which links directly to a PDF that explains this assay.

We have written an explanation of the principle of the assay below, but it is quite long and technical. We would be happy to add this if the editor feels it is necessary.

Briefly, whole blood is mixed with ruthenium labelled folate binding protein which forms a complex with folate in the sample. Streptadivin-coated microparticles and folate labelled with biotin are added to the mixture and bind to the unbound sites of the ruthenium labelled folate binding protein forming a complex. The microparticles are magnetically captured onto a face of an electrode and the unbound folate removed. Applying a voltage to the electrode induces chemiluminescent emission which is measured. The amount of emission is inversely proportional to the amount of folate in the sample.

Are there multiple sub-methods in immunoassays? If so, please describe the difference between them and specify which type the AHMS used.

There are many different types of immunoassays, that all use roughly the same principle. The earliest ones used I125 which has now been replaced by non-radioactive methods such as chemiluminescence. Siemens, Roche, and Abbott all have platforms and assays that can measure red blood cell folate. They all suffer from problems with accuracy but moreover there has been no attempt at harmonization. Even within a method/manufacturer software updates and changes to calibrators give different results over time.

These are all important considerations but are beyond the scope of this paper and we do not have results to support or refute the concerns.

The key point is that we used an external accredited pathology service who used the exact same method used in the AHMS.

We have added the following at ‘Line 73’

“Here we compare RBC folate concentrations measured in blood samples collected from women using the immunoassay (as used in the AHMS survey) and the gold standard microbiological assay method.”

and Line 93.

“This method was identical to that used in the AHMS.”

How many sub-methods exist in microbiological assays? And cutoffs for NTD prevention etc.

The reviewer is correct, there is a lot of debate around cut-offs for NTD prevention and methodological concerns (i.e. particularly use of calibrator folic acid versus 5 methyltetrahydrofolate). We acknowledge systematic biases are likely present especially in the early assays. For example, the O’Broin study was done when there were no external controls and assays will always be driven by the choice of calibrator and the accuracy in which it was measured and the serial dilutions that were required. 

This is why we caution in the discussion on over-interpreting the cutoff of O’Broin et al. starting at line 163

There are limitations of our study. Although attaining a red blood cell folate concentration of 906 nmol/l is a threshold recommended by WHO and others, it does not represent a threshold for the maximum prevention of neural tube defects. There is debate around the optimal cutoff for NTD prevention. An RBC folate concentration of 906 nmol/L was the lower bound of the upper quintile of red blood cell folate in the Irish case–control study [7]. We stress that the model for the relationship between RBC folate and NTD risk is continuous and have provided NTD risk estimates using both the immunoassay and microbiological methods (that differ substantially). Any increase in RBC folate would be expected to decrease NTD risk. However, we accept that our estimates of NTD risk based on RBC folate should be interpreted with caution as they are based on data from a different population [7].

Concerns about calibrator and PMID 25073783 and PMID: 21613453

The reviewer is correct to raise these concerns. I discussed this with K Crider at the US CDC, the lead author of PMID 25073783. They recommend 748 nmol/L as the cutoff for NTD prevention when the chloramphenicol resistant bacteria is used and 5MTHF is used as a calibrator. However, this has not been endorsed by the WHO. Moreover, the relationship between RBC folate concentrations determined by 5MTHF and folic acid is not linear so although she is confident in the cut-off she indicates that the difference is not linear and thus we cannot simply adjust the values especially at high and low values.

In response we have added the following to the discussion and added 1 reference recommended by the reviewer and another that summarizes the outcome of the round robin testing in PMID: 21613453 and includes cutoffs. The following has been added at Line 173

Another important consideration is that we used 5-methyltetrahydrofolate as a calibrator, which is consistent with current recommendations, instead of folic acid as was used in earlier studies. It has been suggested that 5-methyltetrahydrofolate gives RBC folate concentrations from the microbiological method that are ~20% lower [17]. As such a lower cutoff of >748 nmol/L for RBC folate has been suggested for NTD protection when 5-methyltetrahydrofolate is used as the calibrator. Krider et al, using data from two studies in China, a large population-based study (n=247 831) and a dose response trial (n=1194), estimated 822 nmol/L (calibrator adjusted) as a cutoff [18]. In our sample, 27% and 34% of women still had RBC folate concentrations below 748 and 822nmol/L, respectively. Importantly, adjusting RBC folate measurements according to the calibrator used would not change conclusions about the high degree of systematic and random error of the immunological method.

Reviewer 2 Report

Tim et al studied the immune assays used by AHMS in the implications of Neural tube defects in co-relation to RBC folate status of Australian women folic acid fortification. The study design for this commentary is well structured and the methods used were appropriate. The current study will shed the light on imperative need to assess the unnecessary or additional feed of folic acid to the non-target populations in Australia.

Minor concern: My concern here is the control used for study. It would be always good to compare the results with a positive control or a negative control, which is missing in this study. All the study subjects were used here is non-pregnant female, it will be good if we see the differences compared with the few study subjects with actual folic acid supplemented (pregnant) subjects. Or authors have to justify/discuss by citing already published results.  

Author Response

We thank reviewer 2 for his/her comments.

My concern here is the control used for study. It would be always good to compare the results with a positive control or a negative control, which is missing in this study. All the study subjects were used here is non-pregnant female, it will be good if we see the differences compared with the few study subjects with actual folic acid supplemented (pregnant) subjects. Or authors have to justify/discuss by citing already published results.

The blood samples for the immuno-assay were analysed exactly as recommended by the ROCHE product insert and control and would be identical to those to lab contracted by the AHMS using the same assay. There really is not a positive control for quantitative assays with a continuous outcome. What we can say is that the Otago lab that conducted our microbiological analyses used serum sample controls provided by the US National Institute of Standards and Technology (NIST) with assigned values at three levels (low, medium and high). All were within acceptable limits. Moreover, the Otago lab participated in a round robin with the US CDC where samples were exchanged, and performed well.

This manuscript is a resubmission of an earlier submission. The following is a list of the peer review reports and author responses from that submission.

Round 1

Reviewer 1 Report

see attachment

Reviewer 2 Report

Succinct but informative manuscript. Important implications for appropriately interpreting health survey data generally and for accurately measuring folate specifically. Please generally review the paper for grammatical errors. My specific comments and questions are below:

Line 45: Please quantify the remark "Many pregnancies...". This statement contributes to the claim of the importance of the research.

Line 60: "...automated clinical analyser". Please provide reference for exact device or kit used if available.

Line 62: Please elaborate on why the microbiological assay is considered the gold standard.

Line 63: Please name the US survey referred to. 

Line 98-100: Please move this information to the participants section. Also, qere the disagreements between the immunoassay and microbiological methods any more prominent in specific categories of participants? For example in specific age ranges or race/ethnicities? Any reason to expect the results to differ by these demographic categories?

Line 105: Please specify the covariates, if any, that were incorporated within the model.

Line 111: You mention the correlation was poor. However, it was statistically significant. What value would you consider to be a strong correlation for this type of a comparison?

Line 112: Please provide guidance regarding how to interpret the Bland-Altman plot as this may be unfamiliar to some readers.

Line 151: "... removing any intra-individual in folate...". Appears to be a missing word after intra-individual.

General statements/questions: You include a paragraph detailing the strengths of the study. Are there any limitations? Is there a scenario where the immunoassay method may be preferred? If so, what would that be and why? Is it possible that both methods are inaccurately measuring RBC folate?